# ON CHOICE OF LOSS FUNCTIONS FOR NEURAL CONTROL BARRIER CERTIFICATES

## ABSTRACT

The design of controllers with correctness guarantees is a primary concern for safety-critical control systems. A Control Barrier Certificate (CBC) is a real-valued function over the state space of the system that provides an inductive proof of the existence of a safe controller. Recently, neural networks have been successfully deployed for data-driven learning of control barrier certificates. These approaches encode the conditions for the existence of a CBC using a rectified linear unit (ReLU) loss function. The resulting encoding, while sound, tends to be conservative, which results in slower training and limits scalability to large, complex systems. Can altering the loss function alleviate some of the problems associated with ReLU loss and lead to faster learning?

This paper proposes a novel encoding with a Mean Squared Error (MSE) loss function, which allows for more scalable and efficient training, while addressing some of the theoretical limitations of previous methods. The proposed approach derives a validity condition based on Lipschitz continuity to formally characterize safety guarantees, eliminating the need for a post-hoc verification. The effectiveness of the proposed loss functions is demonstrated through six case studies curated from the existing state of the art. Our results provide a compelling argument for exploring alternative loss function choices as a novel approach to optimizing the design of control barrier certificates.

## 1 INTRODUCTION

Recent advances in deep learning have accelerated the integration of autonomous systems into various safety-critical areas of everyday life, including self-driving cars, robotic manipulators, and personalized implantable medical devices. Consequently, even a minor fault in the control logic of these systems can lead to catastrophic consequences, such as loss of human life, severe financial losses, legal liabilities, and damage to infrastructure. In response to this grand challenge, the development of formally certified control methods for autonomous systems has received a considerable research interest in recent years (Xu et al., 2017; Salamati et al., 2024; Zhong et al., 2023; Zhang et al., 2024). Control Barrier Certificates (CBCs) (Ames et al., 2019; Prajna et al., 2007)—and their neural network representations (Dawson et al., 2022; 2023; Liu et al., 2023; Anand & Zamani, 2023; Zhang et al., 2024; Zhao et al., 2021a; Edwards et al., 2024; Qin et al., 2021)—have emerged as leading approach to design a safety controller along with an inductive proof of correctness. This paper focuses on the crucial role that the choice of loss functions plays in the scalable design of safety controllers.

**Neural Control Barrier Certificates.** The key idea behind *control barrier certificates* (CBCs) is that: if one can learn a real-valued function of the state space of a dynamical system such that this function is negative in the initial states, positive in the unsafe states, and, for every state with a non-positive value, there is a control signal choice that allows a transition to another state with a non-positive value, then a feedback control exists that keeps the system safe indefinitely. Traditionally, Sum-of-Squares (SOS) optimization has been used to synthesize such certificates and corresponding controllers (Zhao et al., 2023; Schneeberger et al., 2023; Prajna et al., 2007); however, their application requires human ingenuity in identifying an appropriate template and tends to scale poorly. CBCs parameterized by neural networks—often referred to as Neural Control Barrier Certificates (NCBCs)—have recently gained traction, owing to their universal approximation capabilities, ease

of automation, and the increasing availability of robust tool support (Dawson et al., 2022; Zhang et al., 2024; Liu et al., 2023; Anand & Zamani, 2023). Due to their data-centric approach, NCBCs only provide guarantees over the finite set of data points used during training. Consequently, the resulting controller requires formal verification to provide rigorous guarantees about safety over the entire continuous state space. This verification is typically achieved by framing the problem as a constraint satisfaction task and solving it using Satisfiability Modulo Theories (SMT) solvers, such as Z3 (De Moura & Bjørner, 2008; 2011). The need for such post-hoc verification introduces another weak link in terms of scalability.

**Choice of Loss functions.** The success of deep-learning based approximation depends on a well-designed loss function to ensure that the model learns the correct objective, converges efficiently, and generalizes well to unseen data (Ma et al., 2021; Li et al., 2018). Following Zhao et al. (2020), the majority of work on NCBC (Anand & Zamani, 2023; Abate et al., 2020; Edwards et al., 2024; Zhao et al., 2021a; Žikelić et al., 2024) encodes the control barrier conditions using a ReLU function ($x \in \mathbb{R} \mapsto \max(x, 0)$). The ReLU loss function is straightforward to encode and provides a natural termination condition for training, as training stops when the loss reaches zero. However, ReLU has some fundamental disadvantages (Goodfellow et al., 2016), such as having a zero Hessian everywhere (which hampers interpretability (Torop et al., 2024)) and the instability of its derivative around the global minimum (which affects convergence and robustness). Moreover, prior work (Anand & Zamani, 2023) have demonstrated that using this loss function results in large Lipschitz constants of the trained networks (and consequently the resulting controllers). In practice, small Lipschitz constants for controllers are desirable to ensure more robust control (Chen, 2013). Moreover, barrier certificates with small Lipschitz constants are preferable due to their robustness with respect to small perturbations in the model of a dynamical system (resulting from mechanical wear and tear or changes in operating conditions), thereby improving the applicability and transferability of the resulting guarantees. Can altering the loss function alleviate some of the problems associated with ReLU loss and lead to faster learning?

**Mean Squared Error (MSE) loss.** Mean Squared Error (MSE) is a popular choice (Goodfellow et al., 2016) for loss functions in regression problems due to its strong convergence guarantees (Allen-Zhu et al., 2019; Cheridito et al., 2022). We investigate the suitability of MSE loss functions for NCBCs by posing the following research questions:

**RQ1** Can MSE effectively encode the conditions of neural control barrier certificates?

**RQ2** Can an MSE-based loss function support intuitive termination checks?

**RQ3** Current methods typically fail to scale to more parameterized neural networks and high-dimensional systems. To what extent do MSE loss functions alleviate this drawback?

**RQ4** Small Lipschitz constants are desirable for 1) interpretability, 2) robustness of training, 3) robustness of the resulting controller, and 4) transferability of the resulting guarantees. How do MSE-based NCBCs compare to ReLU-based NCBCs in this regard?

**Contributions.** Our contributions in addressing these research questions are summarized below.

**RQ1** We reformulate, in Section 3, the traditional CBC conditions using MSE loss functions. By leveraging MSE loss, we enable smoother gradients, thereby improving the stability and convergence of neural network training.

**RQ2** In Section 4, we leverage mild Lipschitz continuity assumptions on the system to establish certain *validity conditions* (Theorem 8) for the resulting network, which, when satisfied, allow us to terminate training and provide safety guarantees over the entire state space. This approach eliminates the need for post-hoc verification, thereby improving the scalability of the overall method.

**RQ3-4** In Section 5, we experimentally address these questions by deploying our approach on six case studies from state-of-the-art literature (Anand & Zamani, 2023; Edwards et al., 2024; Zhang et al., 2024; Zhao et al., 2023). Our results show that, compared to existing work, our approach offers greater scalability in terms of system dimensions and neural network architecture, and is able to find formally correct NCBCs faster than current methods. Furthermore, our experiments demonstrate that our approach produces barrier certificates and

controllers with smaller Lipschitz constants, which significantly facilitates the verification process, compared to the state of the art using ReLU loss.

## 2 PROBLEM FORMULATION

As usual, we denote the set of reals, non-negative reals, and positive reals by $\mathbb{R}$, $\mathbb{R}_{\geq 0}$, and $\mathbb{R}_{>0}$, respectively. For sets $A$ and $B$, we write $A \setminus B$ and $A \times B$ for their difference and Cartesian product, respetively. We write $|A|$ for the cardinality of the set $A$. We consider $n$-dimensional Euclidean space $\mathbb{R}^n$ equipped with infinity norm $\|\cdot\|$, defined as $\|x - y\| := \max_{1 \leq i \leq n} |x_i - y_i|$ and Euclidean norm as $\|x - y\|_2 := \sqrt{\sum_{i=1}^{n} (x_i - y_i)^2}$, where $x = (x_1, x_2, \ldots, x_n), y = (y_1, y_2, \ldots, y_n)$ belong to $\mathbb{R}^n$. We denote the rectified linear unit function by $\mathsf{ReLU}(x) := \max(x, 0)$, and mean squared error function by $\mathsf{MSE}(x, y) = \frac{1}{n} \sum_{i}^{n} (x_i - y_i)^2$.

### 2.1 CONTROL BARRIER CERTIFICATES

Systems studied in this paper are modeled as a discrete-time control system (dtCS), defined as follows.

**Definition 1** (Discrete-Time Control System). *A discrete-time control system (dtCS) is a tuple $\mathfrak{S} := (\mathcal{X}, \mathcal{X}_0, U, f)$, where $\mathcal{X} \subseteq \mathbb{R}^n$ represents the continuous state set, $\mathcal{X}_0 \subseteq \mathcal{X}$ is the initial state set, and $U \subseteq \mathbb{R}^m$ is the set of inputs. Furthermore, $f : \mathcal{X} \times U \to \mathcal{X}$ is the state transition function. The evolution of the system under an input sequence $u = \langle u(1), u(2), \ldots \rangle$ is given by*

$$\mathfrak{S} : x(t+1) = f(x(t), u(t)). \tag{1}$$

We assume that sets $\mathcal{X}$, and $U$ are bounded, and the map $f$ is unknown but can be simulated via a black-box model, and $f$ is Lipschitz continuous, as stated in the following assumption.

**Assumption 2** (Lipschitz Continuity). *For a given dtCS $\mathfrak{S} = (\mathcal{X}, \mathcal{X}_0, U, f)$, we assume that $f$ is Lipschitz continuous, i.e., there exists (Lipschitz) constants $\mathcal{L}_u, \mathcal{L}_x \in \mathbb{R}_{\geq 0}$ such that for all $x, x' \in \mathcal{X}$, and $u, u' \in U$, we have*

$$\|f(x, u) - f(x', u')\| \leq \mathcal{L}_x \|x - x'\| + \mathcal{L}_u \|u - u'\|. \tag{2}$$

A dtCS $\mathfrak{S} = (\mathcal{X}, \mathcal{X}_0, U, f)$ with a feedback controller $k(x) : \mathcal{X} \to U$ is *safe* against a set of unsafe states $\mathcal{X}_u \subseteq \mathcal{X}$ if, for every trace of the system starting from $\mathcal{X}_0$ under inputs provided by controller $k$, it never reaches $\mathcal{X}_u$. The main safe control problem studied here is formalized below.

**Problem 3** (Safe Controller Synthesis). *Given a dtCS $\mathfrak{S} = (\mathcal{X}, \mathcal{X}_0, U, f)$, find a feedback controller $k : \mathcal{X} \to U$ such that $\mathfrak{S}$ is safe with respect to initial set of states $\mathcal{X}_0 \subseteq \mathcal{X}$ and unsafe set $\mathcal{X}_u \subseteq \mathcal{X}$, i.e., for every trace $\langle x(0), x(1), \ldots \rangle$, where $x(t+1) = f(x(t), k(x(t)))$, and $x(0) \in \mathcal{X}_0$, we have that $x(t) \notin \mathcal{X}_u$ for all $t \in \mathbb{N}$.*

We employ the following notion of control barrier certificates (CBCs) (Anand et al., 2022) which provides sufficient conditions for ensuring safety.

**Definition 4** (Control Barrier Certificates). *A function $B : \mathcal{X} \to \mathbb{R}$ is called a control barrier certificate (CBC) for $\mathfrak{S} = (\mathcal{X}, \mathcal{X}_0, U, f)$ with respect to initial set of states $\mathcal{X}_0 \subseteq \mathcal{X}$ and unsafe set $\mathcal{X}_u \subseteq \mathcal{X}$ if there exists a controller $k : \mathcal{X} \to U$ such that, for some $\eta \in \mathbb{R}_{\geq 0}$, we have:*

$$B(x) \leq -\eta, \qquad \text{for all } x \in \mathcal{X}_0, \tag{3}$$

$$B(x) > \eta, \qquad \text{for all } x \in \mathcal{X}_u, \text{ and} \tag{4}$$

$$(B(x) \leq 0) \implies (B(f(x, k(x))) \leq 0), \qquad \text{for all } x \in \mathcal{X}. \tag{5}$$

We borrow the next theoretical result from Anand et al. (2022), which outlines the efficacy of CBCs.

**Theorem 5** (Control Barrier Functions Imply Safety). *Consider a dtCS $\mathfrak{S} = (\mathcal{X}, \mathcal{X}_0, U, f)$, and unsafe set of states $\mathcal{X}_u \subseteq \mathcal{X}$. A control barrier certificate that satisfies conditions (3)-(5), guarantees that the system $\mathfrak{S}$, equipped with CBC's controller, starting from any $x \in \mathcal{X}_0$, will never reach $\mathcal{X}_u$ (Anand et al., 2022).*

## 2.2 Neural Control Barrier Certificates

Neural networks, being universal approximators (Hornik et al., 1989), are able to represent any Borel-measurable function based on input-output data. Consider a neural network $F$ with $k$ fully-connected layers where each layer $i$ is characterized with a weight matrix $W_i$ and a bias vector $b_i$ of appropriate size and is followed by an activation function. Such a network can be viewed as a function $F : \mathbb{R}^{n_i} \to \mathbb{R}^{n_o}$. Given $y_0 \in \mathbb{R}^{n_i}$, a network will computes its output $y_k \in \mathbb{R}^{n_o}$ as:

$$y_1 = \sigma(W_1 y_0 + b_1), y_2 = \sigma(W_2 y_1 + b_2), , \cdots, y_k = \sigma(W_k y_{k-1} + b_k).$$

We call $y_{i-1}$ and $y_i$, for $i \in \{1, \ldots, k\}$, the input and output of the $i$-th layer, respectively, and $\sigma$ is the activation function. One observes that neural networks with ReLU ($\sigma(x) = \max(0, x)$) activations describe local Lipschitz continuous functions, with Lipschitz constant $\mathcal{L}_F \in \mathbb{R}_{\geq 0}$, in the sense that for all $x_1', x_2' \in \mathbb{R}^{n_i}$, the following condition holds:

$$\|F(x_1') - F(x_2')\| \leq \mathcal{L}_F \|x_1' - x_2'\|. \tag{6}$$

Moreover, an upper bound on the Lipschitz constant of a neural network with ReLU activations can be obtained using the spectral norm (Combettes & Pesquet, 2020). While tighter Lipschitz upper bounds for neural networks have been extensively studied (Fazlyab et al., 2019; Pauli et al., 2021; Prach & Lampert, 2022; Meunier et al., 2022; Wang et al., 2024; Araujo et al., 2023), we observed that these methods are either too restrictive or introduce significant computational complexity during the training process, as demonstrated in our experiments. The spectral norm approach strikes a good balance: it provides a much tighter bound than the trivial upper bound while remaining computationally efficient.

We focus on how to train neural networks to act as control barrier certificates. To this end, we first introduce the construction of the training set. To do so, we cover the set $\mathcal{X}$ with finitely many disjoint hypercubes $X_1, X_2, \ldots, X_M$, by picking a *discretization parameter* $\epsilon > 0$ such that:

$$\|x - x_i\| \leq \frac{\epsilon}{2}, \text{ for all } x \in X_i, \tag{7}$$

where $x_i$ is the center of hypercube $X_i$, $i \in \{1, \ldots, M\}$. Accordingly, we pick the centers of these hypercubes as sample points, and denote the set of all sample points by $\mathcal{X}_d := \{x_1, \ldots, x_M\}$.

We are ready to propose our notion of neural control barrier certificates.

**Definition 6** (Neural Control Barrier Certificates). *Consider a dtCS $\mathfrak{S} = (\mathcal{X}, \mathcal{X}_0, U, f)$, constants $\epsilon, \eta, \gamma \in \mathbb{R}_{>0}$ such that $\gamma \leq \eta$, the unsafe set $\mathcal{X}_u \subseteq \mathcal{X}$, and neural networks $B : \mathcal{X} \to \mathbb{R}$ and $k : \mathcal{X} \to U$. We proclaim that $B$ along with $k$ is a neural control barrier certificate, if the following conditions hold:*

$$B(x) \leq -\eta, \qquad \text{for all } x \in \mathcal{X}_0 \cap \mathcal{X}_d, \tag{8}$$
$$B(x) > \eta, \qquad \text{for all } x \in \mathcal{X}_u \cap \mathcal{X}_d, \text{ and} \tag{9}$$
$$(B(x) \leq \gamma) \implies (B(f(x, k(x))) \leq -\eta), \qquad \text{for all } x \in \mathcal{X} \cap \mathcal{X}_d, \tag{10}$$

*where $\mathcal{X}_d$ is constructed according to (7), with discretization parameter $\epsilon$.*

In previous works, condition (5) is often replaced with $B(f(x, k(x))) - B(x) \leq -\eta$, which requires the barrier certificate to decrease as the system evolves. Although this is a more conservative condition, it simplifies the verification process (Nejati & Zamani, 2023; Nejati et al., 2023; Anand & Zamani, 2023). Additionally, it is typically assumed that this decreasing condition must hold over the entire state space, a restrictive assumption since some states may not be reachable, yet are still required to satisfy this condition. We tackle this by employing an implication-based approach. We also set $\gamma = \mathcal{L}_B \frac{\epsilon}{2}$, where $\epsilon$ is the discretization parameter, to address this limitation while still ensuring safety guarantees.

Current methods for training neural networks, to act as control barrier certificates for a dtCS $\mathfrak{S} = (\mathcal{X}, \mathcal{X}_0, U, f)$, utilize $L_{\mathsf{ReLU}} := L_1 + L_2 + L_3$ as loss function, where

$$L_1 := \mathsf{ReLU}(B(x), -\eta), \quad \text{for all } x \in \mathcal{X}_d \cap \mathcal{X}_0,$$
$$L_2 := \mathsf{ReLU}(B(x), \eta), \quad \text{for all } x \in \mathcal{X}_d \cap \mathcal{X}_u,$$
$$L_3 := \mathsf{ReLU}(B(f(x, k(x))) - B(x), -\eta), \quad \text{for all } x \in \mathcal{X}_d \setminus \mathcal{X}_u,$$

which $L_1, L_2$ and $L_3$ correspond to conditions (3) to (5), respectively. The advantage of using ReLU is that one can stop the training when loss reaches zero, however, from both theoretical and implementation standpoint, this loss leads to unstable training. Thus, algorithms that use ReLU do not scale well with regards to the dimension of a system and number of parameters of neural networks. To alleviate this drawback, we utilize mean squared error (MSE) loss, which offers guarantees of convergence for over-parameterized neural networks (Allen-Zhu et al., 2019; Cheridito et al., 2022).

## 3 NEURAL CONTROL BARRIER CERTIFICATES WITH MSE LOSS

We propose an alternative approach by replacing the ReLU activation function with a MSE-based formulation for constructing Neural Control Barrier Certificates. The motivation behind this substitution is to exploit the smooth and continuous nature of MSE, which can lead to more efficient gradient-based optimization and improve the overall performance and robustness of the system equipped with the designed controller.

We train $B(x)$ and $k(x)$ with the following loss function $L_{\mathsf{MSE}} = L_1 + L_2 + L_3$, where

$$L_1 := \mathsf{MSE}(B(x), -\eta), \quad \text{for all } x \in \mathcal{X}_d \cap \mathcal{X}_0, \tag{11}$$

$$L_2 := \mathsf{MSE}(B(x), \eta), \quad \text{for all } x \in \mathcal{X}_d \cap \mathcal{X}_u, \tag{12}$$

$$L_3 := \mathsf{MSE}(B(f(x, k(x))), -\eta), \quad \text{for all } x \in \mathcal{X}_d \setminus \mathcal{X}_u, \text{ such that } B(x) \leq \gamma, \tag{13}$$

for an $\eta \in \mathbb{R}_{>0}$, which is a design parameter. Specifically, $L_1, L_2$ and $L_3$ encode conditions (3) to (5) of control barrier certificate, respectively. Additionally, we train the network $k(x)$ with $L_3$. Note that this loss depends on both networks, hence, training both $B(x)$ and $k(x)$ requires dealing with the moving target problem (Mnih et al., 2015). To remedy this problem, we fix $B$ for a predefined number of iterations for loss $L_3$.

To motivate the use of MSE theoretically, we present the following simple example. Consider the scalar system $\mathfrak{S} = (\mathcal{X}, \mathcal{X}_0, U, f)$, where the dynamics are given by $f(x) = \frac{x}{2}$, with $\mathcal{X} = [-10, 10]$, the initial set $\mathcal{X}_0 = [3, 4]$, and the unsafe set $\mathcal{X}_u = [-10, 0)$. Since this system is positive, we have $x(t) \geq 0$ and thus the system is safe, for every $t \in \mathbb{N}$. Consider a barrier certificate given by a linear neural network $B(x) = Mx$ with the Lipschitz constant $\mathcal{L}_B = |M|$. When using ReLU loss, any non-positive value of $M$ leads to a loss of 0. However, with MSE loss, non-positive $M$ with large absolute value—which corresponds to a larger Lipschitz constant for the barrier certificate—results in a larger loss. In this sense, MSE promotes barrier certificates with smaller Lipschitz constants, leading to effectively smoother barrier functions.

Algorithm 1 summarizes our training framework. First, we construct the training data set $\mathcal{X}_d$, and networks are initialized. Then training begins with $L_{\mathsf{MSE}}$. During training, we check for the smallest value of $\eta$ that satisfies conditions (8)-(10), and conditions (14)-(15), if an admissible $\eta$ is found, then training concludes, otherwise training continues. We have also added small regularizers to both networks $B$ and $k$, to encourage both networks to have a small Lipschitz constant (Goodfellow et al., 2016).

Note that a neural control barrier certificate is not necessarily a valid control barrier certificate as in Definition (4), since the training is performed only over a finite set of data. To address this issue, we propose the following validity conditions, which will be utilized to prove that a neural control barrier certificate satisfies conditions of Definition (4), *i.e.,* extend guarantees for training samples to unseen samples.

**Assumption 7** (Validity Conditions). *Consider a dtCS $\mathfrak{S} = (\mathcal{X}, \mathcal{X}_0, U, f)$, and two neural networks $B(x) : \mathcal{X} \to \mathbb{R}$ and $k(x)\mathcal{X} \to U$, with* ReLU *activations that satisfy (8) to (10) for $\mathcal{X}_d$ constructed according to (7). We assume the following validity conditions:*

$$\mathcal{L}_B(\mathcal{L}_x + \mathcal{L}_u\mathcal{L}_k)\frac{\epsilon}{2} - \eta \leq 0, \tag{14}$$

$$\mathcal{L}_B\frac{\epsilon}{2} - \eta \leq 0, \tag{15}$$

*where $\mathcal{L}_B$ and $\mathcal{L}_k$ are Lipschitz constants of networks $B$ and $k$, respectively, and $\mathcal{L}_x$ and $\mathcal{L}_u$ are Lipschitz constants of $\mathfrak{S}$ as defined in (2), and $\epsilon$ is the discretization parameter, and $\eta \in \mathbb{R}_{>0}$ is a user-defined robustness parameter.*

---

**Algorithm 1** Algorithm for Training a Neural Control Barrier Certificate with Formal Guarantee

---

**Input:** Sets $\mathcal{X}_0, \mathcal{X}, U$ for a dtCS $\mathfrak{S}$, respectively, as in Definition (1); discretization parameters $\epsilon$ for the set $\mathcal{X}$ as in (7); robustness parameters $\eta \in \mathbb{R}_{>0}$ as in Definition (6); $\mathcal{L}_x, \mathcal{L}_u$ as introduced in Assumption (2); the number of iterations $N$ for fixing network $B$; the architecture of the neural networks $B$ and $k$; and maximum number of iterations $N_{\max}$.

**Output:** Neural networks $B$ and $k$.

   Construct the training data set $\mathcal{X}_d$ according to 7.

   Initialize networks $B$ and $k$ (Goodfellow et al., 2016).

   $\mathcal{L}_B \leftarrow$ Upper bound of Lipschitz constant of $B$ (Combettes & Pesquet, 2020).

   $\mathcal{L}_k \leftarrow$ Upper bound of Lipschitz constant of $k$ (Combettes & Pesquet, 2020).

   $i \leftarrow 0$

   **while** Conditions (8)-(10) and conditions (14)-(15) are not satisfied and $i \leq N_{\max}$ **do**

      **if** i=nN **then**

         $B_3 = B$.

      **end if**

      Train $B$ with loss $L_{\mathsf{MSE}} = L_1 + L_2 + L_3$, with $L_1, L_2,$ and $L_3$ as in 11-13, respectively.

      Train $k$ via loss $L_3$ generated from $B_3$.

      $i \leftarrow i + 1$

      $\mathcal{L}_B \leftarrow$ Upper bound of Lipschitz constant of $B$ (Combettes & Pesquet, 2020).

      $\mathcal{L}_k \leftarrow$ Upper bound of Lipschitz constant of $k$ (Combettes & Pesquet, 2020).

   **end while**

   Return $B, k$

---

Lipschitz continuity enables us to extend guarantees from a finite set of training data to the entire state set. Assumption 7 serves as a condition that facilitates this extension. Specifically, it ensures that if a sample point (used during training) satisfies the control barrier certificate conditions, then all points within a neighborhood centered at the sample point with radius $\frac{\epsilon}{2}$ also satisfy those conditions. This approach forms the theoretical foundation needed to bridge the gap between finite data and overall correctness across the entire state set.

Although $\eta$ is user-defined, a CBC does not need to satisfy conditions (8)-(10), and conditions (14)-(15) with that given value. Any positive value that satisfies those conditions (8)-(15) provides formal guarantee of safety.

## 4 PROOF OF CORRECTNESS

In this section, we propose the main theoretical result of our paper, and formally prove that a neural control barrier certificate, synthesized according to Algorithm 1, conditioned on its termination, is in fact a control barrier certificate, *i.e.,* it satisfies conditions (3)-(5), and can be deployed to solve Problem (3).

**Theorem 8** (Validity Condition Imply Formal Correctness). *Consider a dtCS* $\mathfrak{S} = (\mathcal{X}, \mathcal{X}_0, U, f)$ *with Lipschitz constants* $\mathcal{L}_x$ *and* $\mathcal{L}_u$ *as in Assumption (2), and a constant* $\epsilon \in \mathbb{R}_{>0}$ *to form* $\mathcal{X}_d$ *as defined in (7). Neural networks* $B : \mathcal{X} \to \mathbb{R}$ *and* $k : \mathcal{X} \to U$ *with Lipschitz constants* $\mathcal{L}_B$ *and* $\mathcal{L}_k$, *respectively, are trained according to Algorithm 1 and represent a neural control barrier certificate. Then* $\mathfrak{S}$ *is safe with respect to the unsafe set* $\mathcal{X}_u \subseteq \mathcal{X}$ *under controller* $k$.

*Proof.* We first prove that condition (5) is satisfied. Consider any $x \in \mathcal{X}$. If $B(x) > 0$, then implication in (5) is trivially satisfied. From now on, we just consider the case that $B(x) \leq 0$. By construction of $\mathcal{X}_d$ as in (7), there exists $x_i \in \mathcal{X}_d$ such that $\|x - x_i\| \leq \frac{\epsilon}{2}$. To obtain an upper bound for $B(x_i)$, we employ Lipschitz continuity:

$$B(x_i) = B(x_i) - B(x) + B(x) \leq \mathcal{L}_B\|x - x_i\| + B(x) \leq \mathcal{L}_B\frac{\epsilon}{2} \leq \gamma.$$

Based on (10), for any $x_i \in \mathcal{X}_d$ such that $B(x_i) \leq \gamma$, one has:

$$B(f(x_i, k(x_i))) \leq -\eta.$$

For all $x \in \mathcal{X}$ such that $\|x - x_i\| \leq \frac{\epsilon}{2}$, and $B(x_i) \leq \gamma$, we have

$$
\begin{aligned}
B(f(x, k(x))) &= B(f(x, k(x))) - B(f(x_i, k(x_i))) + B(f(x_i, k(x_i))) \\
&\leq B(f(x, k(x))) - B(f(x_i, k(x_i))) - \eta \\
&\leq \mathcal{L}_B \|f(x, k(x)) - f(x_i, k(x_i))\| - \eta,
\end{aligned}
$$

where the last inequality follows from Lipschitz continuity of $B$. Moreover:

$$
\begin{aligned}
\mathcal{L}_B \|f(x, k(x)) - f(x_i, k(x_i))\| - \eta &\leq \mathcal{L}_B (\mathcal{L}_x \|x - x_i\| + \mathcal{L}_u \|k(x) - k(x_i)\|) - \eta \\
&\leq \mathcal{L}_B (\mathcal{L}_x + \mathcal{L}_u \mathcal{L}_k) \|x - x_i\| - \eta,
\end{aligned}
$$

which is followed by Assumption 2 and Lipschitz continuity of $k$. According to Algorithm 1, validity condition (14) holds, thus:

$$
B(f(x, k(x))) \leq \mathcal{L}_B (\mathcal{L}_x + \mathcal{L}_u \mathcal{L}_k) \frac{\epsilon}{2} - \eta \leq 0.
$$

Therefore, it follows that condition (10) with validity condition (14) implies condition (5). One could use similar arguments to prove that conditions (3) and (4) hold, however it is omitted here for the sake of brevity. Consequently, a neural control barrier certificate synthesized according to Algorithm 1, is a control barrier certificate as in Definition 4, which guarantees safety of $\mathfrak{S}$ under controller $k$, according to Theorem 5. $\qquad\square$

## 5 Experimental Evaluation

Thus far, we have answered **RQ1** and **RQ2** in previous sections, and here, we aim to address **RQ3** and **RQ4**. We demonstrate the efficacy of our Algorithm with six case studies, two of which are highlighted here. Information regarding the other 4 case studies is found in the Appendix. Table 1 shows a detailed comparison between our method and other state-of-the-art algorithms. We considered methods that 1) provide formal guarantee and 2) train a feedback controller. Among these methods, Anand & Zamani (2023) is model-free, rest require closed-form mathematical expression of map $f$. Moreover, some methods such as Zhang et al. (2024) are for continuous time systems only, however, we discretize systems with forward Euler method (Gottlieb et al., 2001) to compare.

Table 1: Comparison of our proposed method and state-of-the-art. Results showcase our algorithm's independence from the architecture of control barrier certificate, since we do not utilize SMT solvers. We denote the runtime by "NA" when an algorithm fails to converge. Each number, in the architecture column (same architecture for both $B$ and $k$), represents number of neurons for each hidden layer (*i.e.,*10-10-10 refers to a neural network with 3 hidden layers, each consist of 10 neurons), and all networks have ReLU activations.

| Benchmark | Architecture | Edwards et al. (2024) | Anand & Zamani (2023) | Zhao et al. (2021a) | Zhang et al. (2024) | Ours |
|---|---|---|---|---|---|---|
| Spacecraft(6d) | 10-10-10 | 130s | NA | 6000s | 300s | **110s** |
| Spacecraft(6d) | 200-200-200-200 | NA | NA | NA | NA | **92s** |
| Obstacle Avoidance(3d) | 10 | 130s | 3600s | 4000s | **7s** | 120s |
| Obstacle Avoidance(3d) | 200-200-200-200 | NA | NA | NA | NA | **70s** |
| Inverted Pendulum(2d) | 10-10-10 | 250s | 2700s | 2200s | 450s | **130s** |
| Inverted Pendulum(2d) | 200-200-200-200 | NA | NA | NA | NA | **120s** |
| Double Inverted Pendulum(4d) | 200-200-200-200 | NA | NA | NA | NA | **800s** |
| Darboux(2d) | 10 | 50s | 600s | 450s | 8s | **5.8s** |
| Darboux(2d) | 200-200-200-200 | NA | NA | NA | NA | **3.5s** |
| Bicycle Steering(3d) | 10 | 300s | 2800s | 2100s | **20s** | 45s |
| Bicycle Steering(3d) | 200-200-200-200 | NA | NA | NA | NA | **42s** |

### 5.1 Discussion

Zhang et al. (2024) assume that a candidate NCBC is already given, and after verification, they synthesize an admissible controller. Therefore, their method performs well on shallow networks. Moreover, they assume that they have access to exact model of the system (same as Edwards et al. (2024); Zhao et al. (2021a)). On the other hand, we train an NCBC from scratch and assume access

to a black-box representation. The only information that we need from the system is the Lipschitz constants $\mathcal{L}_x$ and $\mathcal{L}_u$, as in Assumption 2. If those constants are unknown, one can leverage sampling based methods to estimate those constants (Wood & Zhang, 1996; Strongin et al., 2019; Calliess, 2017). As shown previously, even with milder assumptions, our method outperforms the existing work and is able to scale to higher dimensional and more complex systems. Furthermore, our method can employ over-parameterized networks to benefit from their representability. Other than scalability, our synthesized controller also has a small Lipschitz constant compared to the rest. This benefit stems from the fact that we have encoded conditions of CBC using MSE loss, which is differentiable in its global minimum (as opposed to ReLU), and comes with convergence guarantees (Allen-Zhu et al., 2019; Cheridito et al., 2022).

We acknowledge that other approaches in the literature provide tighter bounds on the Lipschitz constant of neural networks compared to Combettes & Pesquet (2020), such as Fazlyab et al. (2019); Pauli et al. (2021); Wang et al. (2024); Araujo et al. (2023). However, these methods are computationally expensive. In fact, our numerical experiments indicate that approximately $99\%$ of the training time is spent calculating the Lipschitz constant, with only$1\%$ dedicated to actual training. The spectral norm approach offers a good trade-off: it provides a much tighter bound than the trivial upper bound while remaining efficient to compute. We performed an a posteriori comparison between the spectral norm approach and the method proposed by Fazlyab et al. (2019). Our results show that the spectral norm approach is an order of magnitude faster than Fazlyab et al. (2019), while the upper bound it provides is only $40\%$ to $50\%$ larger than the upper bound obtained by Fazlyab et al. (2019), which can be offset by $\epsilon$ and $\eta$. This finding aligns with the results reported in Fazlyab et al. (2019), particularly in Figure 2a.

## 5.2 EXPERIMENTS SETTING

All of the training is conducted on an Nvidia RTX 4090 GPU coupled with an Intel Core I7 13700k CPU, with 32 GBs of RAM. We utilize Adam optimizer to train neural networks, with a learning rate of $5 \times 10^{-5}$. We have only highlighted pendulum case studies, as they are the most challenging, due to the nonlinearity and dimensionality of systems.

## 5.3 CASE STUDY: INVERTED PENDULUM

We consider a dtCS $\mathfrak{S} = (\mathcal{X}, \mathcal{X}_0, U, f)$ to be an inverted pendulum where $\mathcal{X} = [\frac{-\pi}{4}, \frac{\pi}{4}] \times [\frac{-\pi}{4}, \frac{\pi}{4}]$, $\mathcal{X}_0 = [\frac{-\pi}{12}, \frac{\pi}{12}] \times [\frac{-\pi}{12}, \frac{\pi}{12}]$, and $\mathcal{X}_u = \mathcal{X} \setminus [\frac{-\pi}{6}, \frac{\pi}{6}] \times [\frac{-\pi}{6}, \frac{\pi}{6}]$. The transition function is given by:

$$\begin{bmatrix} \theta(t+1) \\ \omega(t+1) \end{bmatrix} = \begin{bmatrix} \theta(t) + \tau\omega(t) \\ \omega + \frac{g\tau}{l}\sin(\theta(t)) + \frac{10\tau}{ml^2}k(x(t)) \end{bmatrix},$$

where $x(t) := [\theta(t), \omega(t)]$, and $\theta$ and $\omega$ are the angular position and velocity, respectively. Moreover, $g = 9.8$ is the gravitational acceleration, and $l = 1$ and $m = 1$ are the length and mass of the pendulum, respectively. Constant $\tau = 0.01$ is the sampling rate, and Lipschitz constants $\mathcal{L}_x = 1.098$, $\mathcal{L}_u = 0.1$, based on Assumption (2). The discretization parameter and input set are $\epsilon = 1.2 * 10^{-3}$, and $U = [-2.5, 2.5]$, respectively. Our method converged with the following parameters: $\mathcal{L}_B = 0.48$, $\mathcal{L}_k = 2.3$, and $\eta = 0.0037$. Anand & Zamani (2023) report a Lipschitz constant of $\mathcal{L}_B = 21$ for barrier certificate and $\mathcal{L}_K = 20$ for its controller. Some state sequences and level sets of CBC are depicted in Figure 1a and Figure 1b, respectively.

## 5.4 CASE STUDY: DOUBLE INVERTED PENDULUM

For our second case study, we consider a double inverted pendulum $\mathfrak{S} = (\mathcal{X}, \mathcal{X}_0, U, f)$, where $f$ is:

$$\begin{bmatrix} \theta_1(t+1) \\ \omega_1(t+1) \\ \theta_2(t+1) \\ \omega_2(t+1) \end{bmatrix} = \begin{bmatrix} \theta_1(t) + \tau\omega_1(t) \\ \omega_1(t) + \tau(g\sin(\theta_1(t)) - \sin(\theta_1(t) - \theta_2(t))\omega_1^2(t)) \\ \theta_2(t) + \tau\omega_2(t) \\ \omega_2(t) + \tau(g\sin(\theta_2(t)) + \sin(\theta_1(t) - \theta_2(t))\omega_2^2(t)) \end{bmatrix} + \tau \begin{bmatrix} 0 & 0 \\ 30 & 0 \\ 0 & 0 \\ 0 & 39 \end{bmatrix} k(x(t)),$$

where $x(t) := [\theta_1(t); \omega_1(t); \theta_2(t); \omega_2(t)] \in [\frac{-\pi}{4}, \frac{\pi}{4}]^4$, $\theta_1$ and $\theta_2$ represent the angular position of the first and the second joint, respectively, and $\omega_1$ and $\omega_2$ are the angular velocity of the first and the second joint,, respectively, and $U = [-3.5, 3.5]^2$ are the inputs applied to the first and second joint,

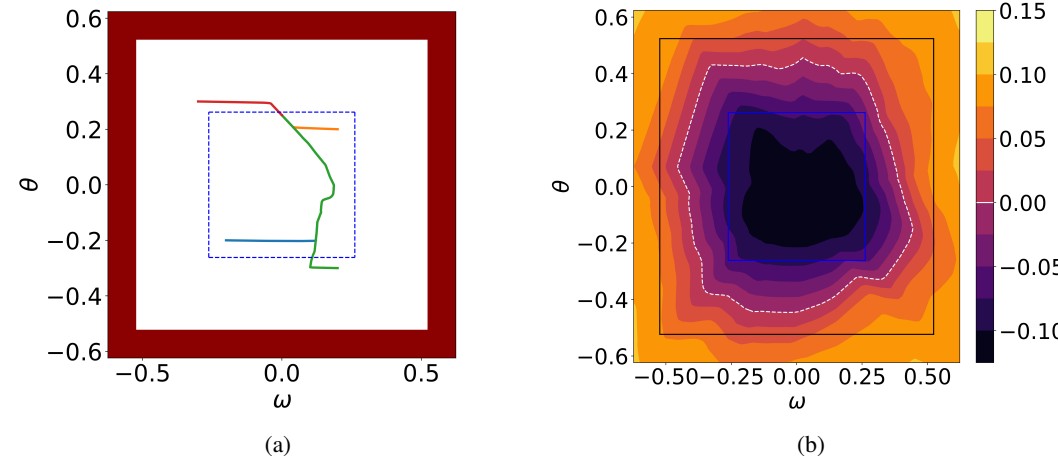

(a)         (b)

Figure 1: Four state sequences of the inverted pendulum are depicted in Figure 1a, starting from different initial conditions; Dotted blue lines indicate the initial set, and red areas depict the unsafe set. Level set of NCBC for the inverted pendulum are depicted in Figure 1b, dotted white, blue, and black lines show the zero-level, the initial set, and the unsafe set of states, respectively.

respectively. Constant $g = 9.8$ is the gravitational acceleration, and Lipschitz constants $\mathcal{L}_x = 1.098$, $\mathcal{L}_u = 0.39$, based on Assumption (2). The initial and unsafe set of states are $\mathcal{X}_0 = [\frac{-\pi}{20}, \frac{\pi}{20}]^4$, $\mathcal{X}_u = \mathcal{X} \setminus [\frac{-\pi}{6}, \frac{\pi}{6}]^4$, respectively, and $\epsilon = 10^{-2}$. Our algorithm converged with the following parameters: $\mathcal{L}_B = 0.17$, $\mathcal{L}_K = 1.8$, and $\eta = 0.00326$. Some trajectories of the system are depicted in Figure 2a and Figure 2b.

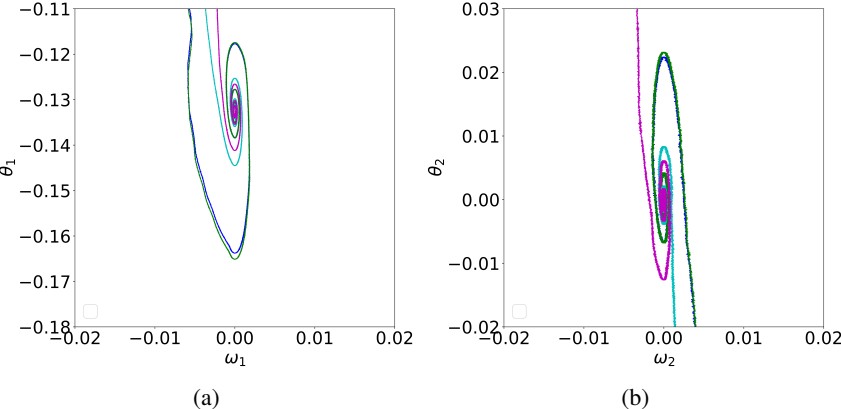

(a)         (b)

Figure 2: Some trajectories of the double inverted pendulum, starting from different initial conditions. Figure 2a and Figure 2b depict the trajectories for the first and second joint, respectively.

## 6 RELATED WORK

**Barrier Certificates.** Prajna & Jadbabaie (2004) first introduced the notion of barrier certificates, whose level sets provide over-approximations of the reachable sets of systems. CBCs emerged as a promising approach to synthesize safe controllers (Ames et al., 2019; Dai & Permenter, 2023; Xiao & Belta, 2019; Clark, 2021; Jagtap et al., 2020). Traditionally, Sum-of-Squars (SOS) optimization is deployed to synthesize such controllers (Zhao et al., 2023; Schneeberger et al., 2023; Prajna et al., 2007). However, these methods require mathematical model of systems and are restricted to polynomial type dynamics only.

**Neural Barrier Certificates.** Zhao et al. (2020) first introduced a notion of neural barrier certificates. They consider a simple, one hidden layer neural network to represent a barrier certificate, and employed Mixed-Integer Linear Programming (MILP) to verify its correctness. Later, Peruffo et al. (2021) utilized SMT solvers to find counter examples to a candidate neural barrier certificate and used those counter examples to train their neural networks. Neural barrier certificates have also been employed for safety verification of hybrid (Zhao et al., 2021b) and stochastic (Mathiesen et al., 2022) systems.

**Neural Control Barrier Certificates.** To tackle the controller synthesize problem, NCBCs have been proposed recently (Dawson et al., 2022; 2023; Liu et al., 2023; Robey et al., 2020; Lindemann et al., 2021; 2024). Existing work utilizes methods such as SMT solvers (Zhao et al., 2020; Edwards et al., 2024; Abate et al., 2020), reachable set verification (Xiang et al., 2018), polynomial approximation (Sha et al., 2021), Lipschitz continuity (Anand & Zamani, 2023), and ReLU networks verification (Katz et al., 2017), to formally verify the correctness of NCBCs. More recently, Zhang et al. (2024) proposed a novel algorithm for exact verification of NCBCs, by considering a given barrier certificate, and synthesizing a controller if that barrier is correct. Almost all of aforementioned algorithms require exact model of a system (with the exception of Anand & Zamani (2023)), and utilize SMT solvers; These solvers cannot deal with deep neural networks efficiently, as the computational complexity grows exponentially with respect to number of parameters, which restricts the architecture of neural networks.

# 7 CONCLUSION

This paper presents advancements in the synthesis and verification of NCBCs by addressing key limitations in prior works. First, by reformulating traditional CBC conditions using MSE loss functions, we introduced smoother gradients, resulting in more stable and efficient training of neural networks. Second, leveraging Lipschitz continuity assumptions, we established training termination conditions that allow guaranteed safety across the entire state space, eliminating the need for post-hoc verification and enhancing scalability. Finally, through experimental validation on six state-of-the-art case studies, we demonstrated that our method improves scalability in terms of system dimensions and network architecture. Additionally, our approach yields synthesized barrier certificates and controllers with smaller Lipschitz constants, simplifying the verification process, and improves robustness and transferability. Possible future direction is to encode conditions of NCBCs using other losses, and investigate effects of MSE on other neural certificates such as Lyapunov (Chang et al., 2019) and Closure certificates (Nadali et al., 2024), and alleviating the sample complexity with properties of the system, such as monotonicity (Angeli & Sontag, 2003) and mixed-monotonicity (Coogan & Arcak, 2015).

# 8 LIMITATIONS

Although our method is capable of utilizing over-parameterized networks, it still suffers from exponential sample complexity, which limits its applicability to higher-dimensional systems. Furthermore, we use the spectral norm method (Combettes & Pesquet, 2020) to estimate the Lipschitz constant of neural networks. While this approach is more conservative compared to other methods such as (Fazlyab et al., 2019; Wang et al., 2024), it offers the advantage of significantly lower computational complexity.

# 9 REPEATABILITY STATEMENT

We have outlined details of our proposed method, with its hyper parameters, and the hardware it was trained on in experiments' section. We have also included the code for inverted pendulum and double inverted pendulum in supplementary materials, as these two case studies are the most challenging among our experiments.

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

## A APPENDIX

### A.1 EXPERIMENT SETTINGS: DARBOUX

This experiment is borrowed from Zhang et al. (2024), which is verification (system is autonomous, *i.e.,* there are no control inputs) for Darboux system, where $x(t) := [x_1(t); x_2(t)]$ are the state of the system, and its dynamic is defined as

$$\left[\begin{array}{c} x_1(t+1) \\ x_2(t+1) \end{array}\right] = x(t) + \tau \left[\begin{array}{c} x_2(t) + 2x_1(t)x_2(t) \\ -x_1(t) + 2x_1^2(t) - x_2^2(t) \end{array}\right].$$

We define state space, initial state, and unsafe state as $\mathcal{X} : \left\{\mathbf{x} \in \mathbb{R}^2 : x \in [0,2] \times [-2,2]\right\}$, $\mathcal{X}_0 : \left\{\mathbf{x} \in \mathbb{R}^2 : 0 \le x_1 \le 1, 1 \le x_2 \le 2\right\}$ and $\mathcal{X}_u : \left\{\mathbf{x} \in \mathbb{R}^2 : \mathbf{x} \in [-2,-1] \times [-2,2]\right\}$, respectively.

### A.2 EXPERIMENT SETTINGS: OBSTACLE AVOIDANCE

This experiment is borrowed from Zhang et al. (2024). The system state consists of 2-D position and aircraft yaw rate $x(t) := [x_1(t); x_2(t); \psi(t)]$. We let $u$ denote the control input to manipulate yaw rate and define the dynamics as

$$\left[\begin{array}{c} x_1(t+1) \\ x_2(t+1) \\ \psi(t+1) \end{array}\right] = x(t) + \tau \left[\begin{array}{c} v\sin(\psi(t)) \\ v\cos(\psi(t)) \\ 0 \end{array}\right] + \tau \left[\begin{array}{c} 0 \\ 0 \\ u \end{array}\right].$$

We define the state space, initial state space and unsafe state space as $\mathcal{X}$, $\mathcal{X}_0$ and $\mathcal{X}_u$, respectively as

$$\mathcal{X} : \left\{\mathbf{x} \in \mathbb{R}^3 : x_1, x_2, \psi \in [-2,2] \times [-2,2] \times [-2,2]\right\};$$
$$\mathcal{X}_0 : \left\{\mathbf{x} \in \mathbb{R}^3 : -0.1 \le x_1 \le 0.1, -2 \le x_2 \le -1.8, -\pi/6 < \psi < \pi/6\right\};$$
$$\mathcal{X}_u : \left\{\mathbf{x} \in \mathbb{R}^3 : x_1 < -0.5 \text{ or } x_1 > 1.5\right\}.$$

### A.3 EXPERIMENT SETTINGS: SPACECRAFT RENDEZVOUS

This experiment is borrowed from Zhang et al. (2024). The state of the chaser is expressed relative to the target using linearized Clohessy–Wiltshire–Hill equations, with state $x(t) := [p_x(t); p_y(t); p_z(t); v_x(t); v_y(t); v_z(t)]$, control input $u(t) = [u_x(t); u_y(t); u_z(t)]$ and dynamics defined as follows.

$$\left[\begin{array}{c} p_x(t+1) \\ p_y(t+1) \\ p_z(t+1) \\ v_x(t+1) \\ v_y(t+1) \\ v_z(t+1) \end{array}\right] = x(t) + \tau \left[\begin{array}{cccccc} 1 & 0 & 0 & 0 & 0 & 0 \\ 0 & 1 & 0 & 0 & 0 & 0 \\ 0 & 0 & 1 & 0 & 0 & 0 \\ 3n^2 & 0 & 0 & 0 & 2n & 0 \\ 0 & 0 & 0 & -2n & 0 & 0 \\ 0 & 0 & -n^2 & 0 & 0 & 0 \end{array}\right] \left[\begin{array}{c} p_x(t) \\ p_y(t) \\ p_z(t) \\ v_x(t) \\ v_y(t) \\ v_z(t) \end{array}\right]$$

$$+\tau \left[\begin{array}{ccc} 0 & 0 & 0 \\ 0 & 0 & 0 \\ 0 & 0 & 0 \\ 1 & 0 & 0 \\ 0 & 1 & 0 \\ 0 & 0 & 1 \end{array}\right] \left[\begin{array}{c} u_x(t) \\ u_y(t) \\ u_z(t) \end{array}\right].$$

We define the state space and unsafe region as $\mathcal{X}$ and $\mathcal{X}_u$, respectively as

$$\mathcal{X} : \left\{\mathbf{x} \in \mathbb{R}^6 : p, v, \in [-1.5, 1.5] \times [-1.5, 1.5]\right\};$$
$$\mathcal{X}_u : \left\{r > 1.5, \text{ where } r = \sqrt{p_x^2 + p_y^2 + p_z^2}\right\}.$$

Task here is to go to the origin without crossing the boundaries.

## A.4  EXPERIMENT SETTINGS: BICYCLE STEERING

This experiment is borrowed from Zhao et al. (2021a). The control objective is to balance a bicycle. The states of the bicycle are $x(t) := [x_1(t); x_2(t); x_3(t)]$ which denote the tilt angle, the angular velocity of tilt, and the handle bar angle with body respectively, and dynamics defined as

$$
\begin{bmatrix} x_1(t+1) \\ x_1(t+1) \\ x_1(t+1) \end{bmatrix} = x(t) + \tau \begin{bmatrix} x_2(t) \\ c_1(g \sin x_1(t) + \frac{v^2}{b} \cos x_1(t) \tan x_3(t)) \\ 0 \end{bmatrix} + \tau \begin{bmatrix} 0 \\ c_2 \cdot \frac{\cos x_1(t)}{\cos^2 x_3(t)} \\ 1 \end{bmatrix} u(t),
$$

where $u$ is the scalar control input, $m = 20$ is the mass, $l = 1$ is the height, $b = 1$ is the wheel base, $J = \frac{mb^2}{3}$ is the moment of inertia, $v = 10$ is the velocity, $g = 9.8$ is the acceleration of gravity, $a = 0.5$, $c_1 = \frac{ml}{J}$, $c_2 = \frac{amlv}{Jb}$, and

$$
\mathcal{X} : \{x \in \mathbb{R}^3 | -2.2 \le x_1 \le 2.2, -2.2 \le x_2 \le 2.2, -2.2 \le x_3 \le 2.2\};
$$
$$
\mathcal{X}_0 : \{x \in \mathbb{R}^3 | -0.2 \le x_1 \le 0.2, -0.2 \le x_2 \le 0.2, -0.2 \le x_3 \le 0.2\};
$$
$$
\mathcal{X}_u : \mathcal{X} \setminus \{x \in \mathbb{R}^3 | -2 \le x_1 \le 2, -2 \le x_2 \le 2, -2 \le x_3 \le 2\}.
$$

