# OpenReview forum: "On Choice of Loss Functions For Neural Control Barrier Certificates"
_ICLR.cc/2025/Conference — Submitted to ICLR 2025_

### Official Review · Reviewer_mhDd · 2024-10-26

**Soundness:** 3
**Presentation:** 2
**Contribution:** 2
**Rating:** 3
**Confidence:** 4

**Summary:**

The paper proposes a Mean Squared Error (MSE)-based loss function for training Neural Control Barrier Certificates (NCBCs) to enhance safety-critical control. By reformulating CBC conditions with MSE, it achieves smoother gradients and better convergence, eliminating the need for post-hoc verification. Experiments confirm improved scalability, stability, and robustness in multiple control scenarios.

**Strengths:**

The MSE loss function allows for smoother gradients and faster convergence, which enhances scalability and enables the NCBCs to handle higher-dimensional systems and more complex neural network architectures.
By satisfying validity conditions through training, the MSE-based approach removes the need for further verification, simplifying the process and making it more computationally efficient.

**Weaknesses:**

1. The primary contribution is the replacement of the ReLU loss with Mean Squared Error (MSE) for smoother convergence in Neural Control Barrier Certificates (NCBCs). The novelty is low.
2. Algorithm 1 is very confusing and not well-written. B3 = B? What is B3 and why?
3. The presentation is poor.
4. the algorithm relies heavily on Lipschitz constants (isn't that loose?)
5. the attached code only includes the pendulum examples.

**Questions:**

See weakness

Any intuition for Theorem 8? It's just math proofs there, and it's a bit hard for readers to digest.

The verification relies heavily on Lipschitz constants, the upper/lower bounds. From my experience, this Lipschitz bound is usually extremely loose, especially when the input region is not small. Did the authors encounter this issue?

What do sections 5.3 and 5.4 mean? Is it to pick an inverted pendulum as an illustration example?

The experiment settings in Section 5.2 miss many hyperparameters, the code only contains the pendulum examples.

---

> ### Author Response · Authors · 2024-11-20
> **Response to Reviewer mhDd (part 1)**
>
> We are grateful for the reviewer’s constructive suggestions and provide our responses below.
>
> Q1. The primary contribution is the replacement of the ReLU loss with Mean Squared Error (MSE) for smoother convergence in Neural Control Barrier Certificates (NCBCs). The novelty is low.
>
> A1. While we agree that replacing the ReLU loss with MSE loss is one of the key contributions of this paper, we believe the work includes other significant novelties in obtaining neural control barrier certificates. For instance, a major contribution is the use of the implication (1):
>
> \begin{equation}
> (B(x) \leq \gamma) \implies (B(f(x, k(x))) \leq -\eta)
> \end{equation}
>
>
> in place of the traditional inequality (2):
>
> \begin{equation}
>     B(f(x, k(x))) - B(x) \leq -\eta
> \end{equation}
>
>
>
> in the third condition of the barrier certificate. This approach is encoded in the neural barrier certificate with MSE loss while still providing formal guarantees.
>
> We note that the implication (1) is significantly less conservative than the traditional inequality $B(f(x, k(x))) - B(x) \leq -\eta$. The latter requires the system’s evolution to be decreasing with respect to the barrier across the set  $\mathcal{X} \setminus \mathcal{X}_u$ , even though some of these states might not be reachable from the initial set. In contrast, our implication relaxes the inequality in two key ways:
>
>  For non-reachable states satisfying  $B(x) > \gamma$, the implication (1) holds trivially, and such states do not impact training.
>     \item Even for reachable states, i.e., states satisfying $B(x)\leq \gamma$ , the implication (1) is a weaker condition than the traditional condition (2).
>
>
> For more details on the implementation of this implication, we refer the reviewer to Assumption 7 (validity conditions) and Theorem 8.
>
> Furthermore, we have rigorously demonstrated that using MSE loss instead of ReLU loss leads to several critical theoretical results regarding the convergence and stability of training.
>
> Additionally, our extensive numerical experiments indicate that, when using MSE loss, both the barrier certificates and controllers exhibit smaller Lipschitz constants compared to those obtained with ReLU loss. This suggests that training with MSE is less data-intensive than training with ReLU loss.
>
>
> Q2. Algorithm 1 is very confusing and not well-written. B3 = B? What is B3 and why?
>
> A2. The term  $B_3 = B$  is included to address the moving target problem, as mentioned in the paper. By fixing the weights of the barrier certificate while training the controller, we ensure that the target we aim to achieve does not "move." For more details on the moving target problem, we refer the reviewer to Mnih et al., "Human-level control through deep reinforcement learning," Nature, 2015. We will add more elaboration on the issue of the moving target problem in the revised version to clearly motivate the use of $B_3 = B$.
>
> Q3. The presentation is poor.
>
> A3. We have revised the draft and addressed comments from other reviewers to improve clarity. We would greatly appreciate it if the reviewer could provide more specific feedback on why the presentation is considered poor, so we can address these concerns more effectively in the revised version.
>
> Q4. the algorithm relies heavily on Lipschitz constants (isn't that loose?)
>
> A4. If one chooses not to use Lipschitz continuity, the only alternative, to the best of our knowledge, is to employ SMT solvers or MILP optimization to provide formal guarantees. However, these methods require an exact mathematical model of the system and suffer from exponential complexity as the number of neurons in the neural network increases. This challenge is particularly significant for over-parameterized networks, which we use in our experiments due to their desirable universal approximation capabilities. We refer the reviewer to Hornik’s work, “Approximation Capabilities of Multilayer Feedforward Networks," Neural Networks, 1991. As demonstrated in our experiments, SMT solvers fail to provide solutions within a reasonable time frame when applied to over-parameterized networks.
>
>
> While the trivial upper bound of the Lipschitz constant is indeed loose, we employ the approach outlined in "Lipschitz Certificates for Layered Network Structures Driven by Averaged Activation Operators," SIAM Journal on Mathematics of Data Science, by Combettes and Pesquet. This approach provides considerably tighter bounds than the trivial upper bound.
>
>
> Q5. the attached code only includes the pendulum examples.
>
> A5. We have included all case studies in the final submission.

---

> ### Author Response · Authors · 2024-11-20
> **Response to Reviewer mhDd (part 2)**
>
> Q6. Any intuition for Theorem 8? It's just math proofs there, and it's a bit hard for readers to digest.
>
> A6. The intuition behind Assumption 7 and Theorem 8 lies in leveraging Lipschitz continuity to provide formal guarantees.
> Since neural networks are trained on a finite set of data points, it is crucial to establish out-of-sample
> performance guarantees to ensure overall correctness
>
> Lipschitz continuity enables us to extend guarantees from a finite set of training data to the entire state set. Assumption 7 serves as a condition that facilitates this extension. Specifically, it ensures that if a sample point (used during training) satisfies the control barrier certificate conditions, then all points within a neighborhood centered at the sample point with radius $\frac{\epsilon}{2}$ also satisfy those conditions. This approach forms the theoretical foundation needed to bridge the gap between finite data and overall correctness across the entire state set.
>
>
>
> Q7. The verification relies heavily on Lipschitz constants, the upper/lower bounds. From my experience, this Lipschitz bound is usually extremely loose, especially when the input region is not small. Did the authors encounter this issue?
>
> A7. While the trivial upper bound of the Lipschitz constant is indeed loose, we employ the approach outlined in "Lipschitz Certificates for Layered Network Structures Driven by Averaged Activation Operators," SIAM Journal on Mathematics of Data Science, by Combettes and Pesquet, which provides considerably tighter bounds than the trivial upper bound.
>
> We have included the Lipschitz constants of our control barrier certificates in Sections $5.2$ and $5.3$. For the inverted pendulum case study, we obtained $\mathcal{L}_B = 0.47$ and $\mathcal{L}_k = 2.3$ as the Lipschitz constants of the barrier certificate and its corresponding controller, respectively. For the double inverted pendulum case study, we obtained $\mathcal{L}_B = 0.17$ and $\mathcal{L}_k = 1.8$ for the barrier certificate and its corresponding controller, respectively.
>
> Q8. What do sections 5.3 and 5.4 mean? Is it to pick an inverted pendulum as an illustration example?
>
> A8. Yes, Sections 5.3 and 5.4 focus on the two most challenging case studies. Additional details about the other four systems are provided in the appendix.
>
> Q9. The experiment settings in Section 5.2 miss many hyperparameters, the code only contains the pendulum examples.
>
> A9. We did our best to include all hyperparameters (such as Lipschitz constants, $\epsilon$, and $\eta$) in their corresponding subsections. However, it is certainly possible that we may have missed some. Could the reviewer please specify which hyperparameters they believe were omitted?
>
>
> We sincerely thank the reviewer for their detailed analysis and thoughtful feedback on our method. We appreciate the time and effort you have dedicated to reviewing our paper and will incorporate your suggestions to enhance its clarity and quality.
>
> We kindly request that you re-evaluate our paper in light of the clarifications and additional insights we have provided.
>
> Sincerely,
>
> Authors

---

> ### Comment · Reviewer_mhDd · 2024-11-24
>
> I have read the authors' response.

---

> ### Author Response · Authors · 2024-11-27
>
> We appreciate the reviewer’s constructive suggestions, which have helped improve our paper. We have included all experiments in the attached code.
>
> If you have any further questions or concerns, please feel free to reach out.

---

### Official Review · Reviewer_yeAM · 2024-11-02

**Soundness:** 2
**Presentation:** 1
**Contribution:** 3
**Rating:** 5
**Confidence:** 5

**Summary:**

An important problem in control theory is to prove that a system can reach a desired condition while avoiding unsafe conditions.  This problem has been formulated using a variety of methods in the past, including constrained optimization, optimal control and dynamic games, and with the use of control Lyapunov and control barrier functions. The challenge in using control Lyapunov or control barrier functions is in the generation of the function itself:  generally these are hand-tuned functions inspired by the linearization of the system and the constraints, and can be hard to construct, and the result is often quite conservative.  Over the last decade, new methods for generating control Lyapunov and control barrier functions have been developed using neural networks.  The idea has been to use the dynamics model and the barrier function conditions to train a neural-network representation of the function and the resulting control policy.  Early work demonstrated that neural networks could learn these conditions, but failed to provide a proof over the state space of the system.  In more recent work, ReLU neural networks have been used to represent the barrier function, and a proof of its applicability over the state space generated by employing formal verification tools based on Satisfiability Modulo Theory (SMT) and the piecewise linear structure of the network.  The conditions for existence of a control barrier function are encoded in the ReLU loss function, and the proof of applicability of the state space can be slow to obtain through the training of this network, which suffers from the curse of dimensionality, resulting in scalability limitations.

The current paper replaces the ReLU loss function in this last step with a mean squared error (MSE) loss function.  Lipschitz continuity conditions of the system and the neural networks implementing the control barrier function are then sufficient to prove the applicability of the control barrier function over the state space, removing the need for a formal verification procedure using SMT tools.  Dynamic system examples ranging from 2D to 6D demonstrate that this new loss function results in computation of a barrier function and resulting control law faster than previous methods.

**Strengths:**

The key idea here is a good one, and the paper builds on previous work to automate the generation of control barrier functions.  The new MSE-based loss function works better than the ReLU-based loss function featured in previous work, and the paper employs Lipschitz continuity assumptions to generate a proof that the barrier function can be used across a region of states, not just at the sample points used for training.  This speeds up the process because the formal verification used in the ReLU-based loss function work is not needed.
The benchmark examples are adopted from the previous work, and allow a direct comparison with previous work.

**Weaknesses:**

(1) The gridding of the state space into Lipschitz-constant-dependent neighborhoods of training points is a potential downside of this approach -- it seems that it could limit the scalability and remove some of the benefits of using a barrier function.  Could the authors include discussion (beyond the examples shown) of how the training scales with the dimension of the dynamic system, as the number of training points is exponential in this dimension.

(2) The presentation of the paper is quite poor.  First, the paper presents results using a discrete time control system, whereas the original work on which CBCs (Prajna) are based is presented for continuous systems.  This should be clarified and that the results carry over directly should be confirmed.  Second, the ReLU(.,.) and MSE(.,.) functions are not defined.  In Alg 1 L_{MSE} references L_i functions in equations (7)-(9) that do not exist.  Lipschitz continuity of neural networks is not defined.  There are small typos throughout the paper.

(3) There is not much physical intuition given in the paper:  why are the inverted pendula examples harder to verify?  Why is the inverted pendulum example not symmetric wrt theta and thetadot?  What is the dependence on the network architecture, and how were the specific architectures used in the examples chosen?   Also, these systems are small enough that there are analytically-determined barrier functions for at least some of them.  How do the computational results compare?  Finally, how is eta chosen, and is it decreased over the course of training?

**Questions:**

See questions in weaknesses above.

---

> ### Author Response · Authors · 2024-11-20
> **Response to Reviewer yeAM (part 1)**
>
> We are grateful for the reviewer’s constructive suggestions and provide our responses below.
>
> Q1. The gridding of the state space into Lipschitz-constant-dependent neighborhoods of training points is a potential downside of this approach -- it seems that it could limit the scalability and remove some of the benefits of using a barrier function. Could the authors include discussion (beyond the examples shown) of how the training scales with the dimension of the dynamic system, as the number of training points is exponential in this dimension.
>
> A1. We acknowledge the reviewer’s observation that sample complexity is exponential. However, computational complexity is a common challenge for data-driven methods that provide formal guarantees, and this issue is not unique to our approach. To further illustrate this point, here are some examples:
>
>
> Ansaripour et al., "Learning provably stabilizing neural controllers for discrete-time stochastic systems," ATVA 2023.
>
>
> Mathiesen et al., "Safety Certification for Stochastic Systems via Neural Barrier Functions," LCSS 2022.
>
>
> Ehlers, "Formal verification of piece-wise linear feed-forward neural network," ATVA 2017.
>
> Elboher et al., "An abstraction-based framework for neural network verification," CAV 2020.
>
>
> Katz et al., "Reluplex: An efficient SMT solver for
> verifying deep neural networks," CAV 2017.
>
> Chang et al., "Neural Lyapunov Control," NeurIPS 2019.
>
>
> Zhao et al.,"Synthesizing barrier certificates using neural networks," HSCC 2020.
>
> Zhang et al., "Exact Verification of ReLU Neural Control Barrier Functions," NeurIPS 2024.
>
> Even assuming the availability of the system's model, we attempted prior methods using Sum of Squares (SOS) optimization techniques but were unsuccessful in finding polynomials (up to degree 15) for the double inverted pendulum case study, ultimately leading us to abandon increasing the degree of the polynomials further.
>
> That said, we believe our proposed method remains more efficient than many model-based techniques, as evidenced by our experimental results.
>
>
> Q2. The presentation of the paper is quite poor. First, the paper presents results using a discrete time control system, whereas the original work on which CBCs (Prajna) are based is presented for continuous systems. This should be clarified and that the results carry over directly should be confirmed. Second, the ReLU(.,.) and MSE(.,.) functions are not defined. In Alg 1 L_{MSE} references L_i functions in equations (7)-(9) that do not exist. Lipschitz continuity of neural networks is not defined. There are small typos throughout the paper.
>
> A2. The reviewer is correct. We have added a citation to the following paper, which defines control barrier certificates for discrete-time systems (setting $k=1$):
>
> Anand et al., “K-inductive barrier certificates for stochastic systems,” HSCC 2022.
>
>
> Furthermore, we have addressed the reviewer’s concerns by providing clear definitions for the ReLU and MSE functions, correcting references in Algorithm 1, and resolving minor typographical errors throughout the paper.
>
> Q3. There is not much physical intuition given in the paper: why are the inverted pendula examples harder to verify?
>
> A3. Both examples are nonlinear systems that must ensure safety around their unstable equilibrium points.
>
> Q4.  Why is the inverted pendulum example not symmetric wrt theta and thetadot?
>
> A4. We have adopted the inverted pendulum from the following state-of-the-art papers:
>
>
>  Edwards et al., "Fossil 2.0: Formal Certificate Synthesis for the Verification and Control of Dynamical Models," HSCC 2024.
>
>
> Anand and Zamani, "Formally Verified Neural Network Control Barrier Certificates for Unknown Systems," IFAC 2023.
>
>
>  Chang et al., "Neural Lyapunov Control," NeurIPS 2019.
>
>
> The model of the double inverted pendulum was derived using the Euler-Lagrange equations, as outlined in the Appendix A.1 of "Course Notes for MIT 6.832" by Tedrake.
>
> Q5. What is the dependence on the network architecture, and how were the specific architectures used in the examples chosen?
>
> A5. The universal approximation capability of a neural network relies on it being over-parameterized, i.e., sufficiently deep and wide. We refer the reviewer to Hornik’s work, "Approximation Capabilities of Multilayer Feedforward Networks," Neural Networks, 1991. Based on this, we chose an architecture with 4 hidden layers, each containing 200 neurons heuristically.

---

> ### Author Response · Authors · 2024-11-20
> **Response to Reviewer yeAM (part 2)**
>
> Q6. Also, these systems are small enough that there are analytically-determined barrier functions for at least some of them. How do the computational results compare?
>
> A6. We agree with the reviewer that some of our case studies involve relatively small systems. However, these examples were adopted from the literature to allow for direct comparisons. To further demonstrate the advantages of our algorithm, we introduced a new system—the double inverted pendulum—where all previous methods fail to find a control barrier certificate, even when assuming access to the exact mathematical model of the system.
>
> Q7. Finally, how is eta chosen, and is it decreased over the course of training?
>
> A7. Regarding $\eta$, it is a design parameter; however, based on our experiments, it does not play a critical role in training since any admissible $\eta$ provides sufficient conditions for safety. The value of $\eta$ depends on several factors, including the system, the geometry of the initial and unsafe regions, the distance between these sets, and the parameters of the neural networks. Generally, larger values of $\eta$ make it easier to satisfy conditions (14)-(15) with a larger quantization parameter for data collection.
>
> Our algorithm computes $\eta$ at each iteration, and training concludes once it meets the admissibility criteria, specifically the validity conditions outlined in Assumption 7.
>
> We sincerely thank the reviewer for their detailed analysis and thoughtful feedback on our method. We appreciate the time and effort you have dedicated to reviewing our paper and will incorporate your suggestions to enhance its clarity and quality.
>
> We kindly request that you re-evaluate our paper in light of the clarifications and additional insights we have provided.
>
> Sincerely,
>
> Authors

---

> ### Author Response · Authors · 2024-12-02
>
> With the rebuttal deadline approaching today, we kindly remind the reviewer to review our response to your comments.
>
> If any clarifications or additional details are needed to assist with your review, please feel free to let us know.
>
> Thank you for your time and valuable feedback.

---

### Official Review · Reviewer_oQrt · 2024-11-05

**Soundness:** 2
**Presentation:** 3
**Contribution:** 2
**Rating:** 3
**Confidence:** 3

**Summary:**

This paper leverages a Mean Squared Error (MSE) loss function for constructing neural CBF and claims that their approach gives which more scalable and efficient training. The proposed approach uses Lipschitz continuity to provide safety guarantees. Numerical results are provided to justify the effectiveness of the proposed method.

**Strengths:**

1. This paper is quite original, and I haven't seen similar ideas before.

2. Numerical study on six different cases is definitely a plus.

3. The paper is written well, and the idea is quite simple and easy to follow.

4. The idea of using Lipschitz analysis to derive control barrier functions is interesting.

**Weaknesses:**

I am not sure how significant the result is. It seems that the authors use Lipschitz constant to guarantee stability. So the authors are using the spectral norm bounds for Lipschitz constants and then derive certificates based on that? I almost feel that the study is not that complete given the facts that there are so many results on Lipschitz networks which are completely ignored by this work. It seems that one can immediately strengthen the results in this paper by connecting the results to the large body of literature on Lipschitz networks. Some sampled results on Lipschitz networks are provided below, and there are actually many more papers.

Miyato et. al: Spectral normalization for generative adversarial networks. ICLR 2018.

Fazlyab et.al: Efficient and accurate estimation of lipschitz constants for deep neural networks. NeurIPS 2019

Li et. al. Preventing gradient attenuation in Lipschitz constrained convolutional networks. NeurIPS 2019.

Pauli et. al: Training robust neural networks using Lipschitz bounds. IEEE LCSS 2021.

Trockman, A. and Kolter, J. Z. Orthogonalizing convolutional layers with the Cayley transform. ICLR 2021.

Singla, S. and Feizi, S. Skew orthogonal convolutions. ICML 2021.

Meunier et.al: A dynamical system perspective for Lipschitz neural networks. ICML 2022

Xu et. al. Lot: Layer-wise orthogonal training on improving l2 certified robustness. NeurIPS 2022

Prach et. al. Almost-orthogonal layers for efficient general-purpose Lipschitz networks. ECCV 2022.

Araujo et al: A unified algebraic perspective on Lipschitz neural networks. ICLR 2023

Wang, R. and Manchester, I. Direct parameterization of lipschitz-bounded deep networks. ICML 2023.

Wang et al: On the scalability and memory efficiency of semidefinite programs for Lipschitz constant estimation of neural networks. ICLR 2024.

Pauli et. al: Novel quadratic constraints for extending LipSDP beyond slope-restricted activations. ICLR 2024.

Prach et. al: 1-Lipschitz Layers Compared: Memory Speed and Certifiable Robustness. CVPR 2024.


This paper almost completely ignores the large body literature on Lipschitz networks. It seems to me that the proposed approach can be significantly strengthened by connecting to these papers. It is also unclear to me whether the key to the improvement is the MSE loss or just the control of Lipschitz constants.

**Questions:**

1. Are the authors using the spectral norm bounds for Lipschitz constants and then derive certificates based on that? Have the authors considered more advanced methods for Lipschitz continuity?

2. Can the authors justify whether the MSE loss or the Lipschitz continuity lead to the improvements? Maybe some baselines that use advanced Lipschitz continuity properties without using MSE loss can be added for comparison?

---

> ### Author Response · Authors · 2024-11-20
> **Response to Reviewer oQrt**
>
> We are grateful for the reviewer’s constructive suggestions and provide our responses below.
>
> Q1. Are the authors using the spectral norm bounds for Lipschitz constants and then derive certificates based on that? Have the authors considered more advanced methods for Lipschitz continuity?
>
> A1. We would like to point out that our method is concerned with safety, and not stability.
>
> Since we estimate the Lipschitz constant of two neural networks at each iteration during training, an efficient method for estimating it is crucial.
> We agree that there are other approaches in the literature that provide tighter bounds on the Lipschitz constant compared to Combettes and Pesquet, such as Fazlyab et al.,"Efficient and accurate estimation of Lipschitz constants for deep neural networks," NeurIPS 2019 and Pauli et al.,"Novel quadratic constraints for extending LipSDP beyond slope-restricted activation," ICLR 2024. However, these methods are computationally expensive. In fact, our numerical experiments suggest that approximately $99\\%$ of the training time is spent on calculating the Lipschitz constant, with only $1\\%$ dedicated to actual training.
>
> Instead, we compute an upper bound of the Lipschitz constant using the spectral norm. For details, we refer the reviewer to "Lipschitz certificates for layered network structures driven by averaged activation operators," SIAM Journal on Mathematics of Data Science, by Combettes and Pesquet. The spectral norm approach offers a good trade-off: it provides a much tighter bound than the trivial upper bound while remaining efficient to compute.
>
> We performed an a posteriori comparison between the spectral norm approach and the method proposed by Fazlyab et al. Our results show that the spectral norm approach is an order of magnitude faster than Fazlyab et al., while the upper bound it provides is only $ 40\\%$ to $50\\%$ larger than the upper bound obtained by Fazlyab. This finding aligns with the results reported in Fazlyab et al., particularly in Figure 2a.
>
> Q2. Can the authors justify whether the MSE loss or the Lipschitz continuity lead to the improvements? Maybe some baselines that use advanced Lipschitz continuity properties without using MSE loss can be added for comparison?
>
> A2. Justification for using MSE loss:
>
> We would like to highlight that using MSE leads to several theoretical and numerical advantages.
>
>  As it is show in the literature, MSE loss has guarantees of convergence and optimality, see Cheridito et al.,"A proof of convergence for gradient descent in the training of artificial neural networks for constant target functions," Journal of Complexity, 2022.
>
>
>  Our extensive numerical experiments indicate that, when using MSE loss, both the barrier certificates and controllers exhibit smaller Lipschitz constants compared to those obtained with ReLU loss. This suggests that training with MSE is less data-intensive than training with ReLU loss, which is advantageous when scaling to larger problems.
>
> Justification for using Lipschitz continuity:
>
> If one chooses not to use Lipschitz continuity, the only alternative, to the best of our knowledge, is to employ SMT solvers or MILP optimization to provide formal guarantees. However, these methods require an exact mathematical model of the system and suffer from exponential complexity as the number of neurons in the neural network increases. This challenge is particularly significant for over-parameterized networks, which we use in our experiments due to their desirable universal approximation capabilities. We refer the reviewer to Hornik’s work, “Approximation Capabilities of Multilayer Feedforward Networks," Neural Networks, 1991. As demonstrated in our experiments, SMT solvers fail to provide solutions within a reasonable time frame when applied to over-parameterized networks.
>
> Furthermore, Anand and Zamani in “Formally Verified Neural Network Control Barrier Certificates for Unknown Systems," IFAC 2023, utilize the approach proposed by Pauli et al., “Training Robust Neural Networks Using Lipschitz Bounds," LCSS 2021. We have compared our method to theirs, and as shown in Table 1 of our paper, our method demonstrates clear superiority in performance.
>
> We sincerely thank the reviewer for their detailed analysis and thoughtful feedback on our method. We appreciate the time and effort you have dedicated to reviewing our paper and will incorporate your suggestions to enhance its clarity and quality.
>
> We kindly request that you re-evaluate our paper in light of the clarifications and additional insights we have provided.
>
> Sincerely,
>
> Authors

---

> ### Author Response · Authors · 2024-12-02
>
> With the rebuttal deadline approaching today, we kindly remind the reviewer to review our response to your comments.
>
> If any clarifications or additional details are needed to assist with your review, please feel free to let us know.
>
> Thank you for your time and valuable feedback.

---

### Official Review · Reviewer_RLbj · 2024-11-09

**Soundness:** 3
**Presentation:** 3
**Contribution:** 2
**Rating:** 6
**Confidence:** 4

**Summary:**

The authors address the challenging problem of synthesizing a neural network controller and barrier certificate simultaneously which certifies safe behavior of the closed-loop system under unknown dynamics. The authors propose an MSE loss and validity condition which can practically achieve state-of-the-art performance in terms of training time and network parameter scalability over previous methods.

**Strengths:**

- The formulation is practical in the sense that it does not assume much structure at all about the dynamics and control other than they are in state-feedback and uses a weaker definition of CBC than previous papers by not requiring incremental decrease in the level sets.

- The algorithm and verification are technically sound and implementable given the required Lipschitz constants and lead to favorable results over existing approaches in their experimental settings of systems up to 6 dimensions.

**Weaknesses:**

-The paper’s formulation and technical results are similar to that of Nejati 2023.

- The validity of the algorithm depends on obtaining valid upper-bounds for the Lipschitz constants L_x and L_u. This seems to be non-trivial unless they are known analytically.

- I don’t think the MSE loss is well-motivated as a contribution and there is not enough justification other than smoothness. Once a performance objective is added to the loss, the CBC MSE loss does not behave like a constraint and will compete with the performance in possibly undesirable ways (related to my questions below).

- The author’s claim that their method is more scalable to higher-dimensions. I will agree that it is scalable wrt to the network size, however I think there are significant scalability issues in state and control dimension in requiring an epsilon-net (Eq (6) on the state-space and and requiring sampled-based estimates for Lipschitz constants L_x and L_u. Often-times large networks aren’t required for direct state-feedback of 6-dimensional problems and more appropriate for perception-based controllers not currently supported by this framework.

**Questions:**

-I don’t totally understand the motivation of the MSE loss. The ReLU-type loss is meant to mimic a constraint and be active only when the constraint e.g. B(x) <= -\eta is not satisfied. However, it seems that MSE loss tends to push B(x) towards the boundary of the safe level set so that B(x) = -eta. It’s not clear how this will affect the overall landscape of the learned barrier function or why this is desirable besides smoothness of the loss. Typically a smooth performance objective is included in the loss anyways, which would end up competing adversarially with your CBC MSE loss.

- The control problem formulation is kind of strange to me. Algorithm 1 certainly trains a controller to avoid the unsafe region, but what is the controller trying to achieve otherwise? Shouldn’t there be some performance objective such as stability to a fixed point or reaching a terminal set? The setting in the paper would make more sense to me if we are already given a controller and want to learn a barrier function B to certify its behavior rather than learning one from scratch. If a performance objective is added to the algorithm, it might be hard to trade-off that additional loss with your MSE which is always going to be non-zero if B is a continuous function.

- Using spectral normalization to enforce the Lipschitz of B and k is quite conservative. Have you tried using more sophisticated Lipschitz parameterizations (see below)?

Pracht 2022: Almost-Orthogonal Layers for Efficient General-Purpose Lipschitz Networks
Wang 2023: Direct Parameterization of Lipschitz-Bounded Deep Networks

---

> ### Author Response · Authors · 2024-11-20
> **Response to Reviewer RLbj (part 1)**
>
> We are grateful for the reviewer’s constructive suggestions and provide our responses below.
>
> Q1. The paper’s formulation and technical results are similar to that of Nejati 2023
>
> A1. While the problem statements in our paper and Nejati (2023) share similarities, particularly in providing data-driven guarantees, there are significant differences in our approaches. Unlike Nejati (2023), which focuses solely on verification and does not employ neural networks, our method is designed for synthesis. Specifically, we leverage neural networks to represent control barrier functions and their corresponding controllers. Nejati (2023) relies on a fixed template for barrier functions and formulates the problem using a scenario convex program, resulting in probabilistic guarantees. In contrast, our framework ensures 100\% correctness of the control barrier functions and their associated controllers, representing a fundamentally different approach.
>
> Q2. The validity of the algorithm depends on obtaining valid upper-bounds for the Lipschitz constants L_x and L_u. This seems to be non-trivial unless they are known analytically.
>
> A2. We agree with the reviewer that obtaining the Lipschitz constants is non-trivial. However, as mentioned in the paper, sampling-based techniques can be employed to estimate these constants, which is a well-studied field. Established methods include:
>
> Strongin et al.,"Acceleration of global search by implementing
> dual estimates for Lipschitz constant," NUMTA 2019.
>
> Novara et al., "Direct feedback control design for nonlinear systems," Automatica 2013.
>
>
> Calliess et al., "Lazily adapted constant kinky inference for nonparametric regression and model-reference adaptive control," Automatica 2020.
>
>
> Calliess, "Lipschitz optimisation for Lipschitz interpolation," ACC 2017.
>
> González et al., "Batch Bayesian optimization via local penalization," Artificial intelligence and statistics, 2016.
>
> Q3. The author’s claim that their method is more scalable to higher-dimensions. I will agree that it is scalable wrt to the network size, however I think there are significant scalability issues in state and control dimension in requiring an epsilon-net (Eq (6) on the state-space and and requiring sample-based estimates for Lipschitz constants $L_x$ and $L_u$. Often-times large networks aren’t required for direct state-feedback of 6-dimensional problems and more appropriate for perception-based controllers not currently supported by this framework.
>
> A3. We agree with the reviewer’s point regarding the exponential sample complexity in requiring an $\epsilon$-net. As mentioned in the conclusion, our future work focuses on addressing this challenge by leveraging system properties such as monotonicity, mixed monotonicity, and compositionality techniques. It is important to note that sample complexity is a common challenge in methods providing formal guarantees and is not unique to our approach. To further illustrate this, here are some examples:
>
> Ansaripour et al., "Learning provably stabilizing neural controllers for discrete-time stochastic systems," ATVA 2023.
>
> Mathiesen et al., "Safety Certification for Stochastic Systems via Neural Barrier Functions," LCSS 2022.
>
>
> Ehlers, "Formal verification of piece-wise linear feed-forward neural network," ATVA 2017.
>
> Elboher et al., "An abstraction-based framework for neural network verification," CAV 2020.
>
>
> Katz et al., "Reluplex: An efficient SMT solver for
> verifying deep neural networks," CAV 2017.
>
> Chang et al., "Neural Lyapunov Control," NeurIPS 2019.
>
>
> Zhao et al.,"Synthesizing barrier certificates using neural networks," HSCC 2020.
>
> Zhang et al., "Exact Verification of ReLU Neural Control Barrier Functions," NeurIPS 2024.
>
>
> Even assuming the availability of the system's model, we attempted prior methods using Sum of Squares (SOS) optimization techniques but were unsuccessful in finding polynomials (up to degree 15) for the double inverted pendulum case study, ultimately leading us to abandon increasing the degree of the polynomials further.
>
>
>  That said, we believe our proposed method is still more efficient than many model-based techniques, as demonstrated in our experiments.

---

> ### Author Response · Authors · 2024-11-20
> **Response to Reviewer RLbj (part 2)**
>
> Q4. I don’t totally understand the motivation of the MSE loss. The ReLU-type loss is meant to mimic a constraint and be active only when the constraint e.g. B(x) <= -\eta is not satisfied. However, it seems that MSE loss tends to push B(x) towards the boundary of the safe level set so that B(x) = -eta. It’s not clear how this will affect the overall landscape of the learned barrier function or why this is desirable besides smoothness of the loss. Typically a smooth performance objective is included in the loss anyways, which would end up competing adversarially with your CBC MSE loss.
>
> A4. We observed that the MSE loss primarily pushes the initial set of states to $-\eta$ and the unsafe set to $\eta$, as these values are directly incorporated into the loss function.
>
> We would like to emphasize that using MSE loss provides several theoretical and numerical advantages:
>
>  Convergence and Optimality Guarantees: As shown in the literature, MSE loss has established guarantees of convergence and optimality. For instance, Cheridito et al., in"A Proof of Convergence for Gradient Descent in the Training of Artificial Neural Networks for Constant Target Functions," Journal of Complexity, 2022,  highlight its robustness in achieving these guarantees.
>
>   Numerical Benefits: Our extensive numerical experiments indicate that the barrier certificates and controllers trained using MSE loss exhibit smaller Lipschitz constants compared to those trained with ReLU loss. This suggests that training with MSE loss is less data-intensive, which is advantageous when scaling to larger problems.
>
> Q5. The control problem formulation is kind of strange to me. Algorithm 1 certainly trains a controller to avoid the unsafe region, but what is the controller trying to achieve otherwise? Shouldn’t there be some performance objective such as stability to a fixed point or reaching a terminal set? The setting in the paper would make more sense to me if we are already given a controller and want to learn a barrier function B to certify its behavior rather than learning one from scratch. If a performance objective is added to the algorithm, it might be hard to trade-off that additional loss with your MSE which is always going to be non-zero if B is a continuous function.
>
> A5. Our experiments show that while training a controller for safety, the system equipped with it tends to reach a fixed point, as demonstrated in our results. Performance objectives can be incorporated into the loss function; however, overly restrictive metrics may prevent convergence.
>
> Moreover, achieving a zero loss is not necessary to establish formal guarantees, as we have outlined in our paper.

---

> ### Author Response · Authors · 2024-11-20
> **Response to Reviewer RLbj (part 3)**
>
> Q6. Using spectral normalization to enforce the Lipschitz of B and k is quite conservative. Have you tried using more sophisticated Lipschitz parameterizations (see below)?
>
>
>
> Pracht 2022: Almost-Orthogonal Layers for Efficient General-Purpose Lipschitz Networks Wang 2023: Direct Parameterization of Lipschitz-Bounded Deep Networks
>
> A6. We do not utilize spectral normalization, nor do we enforce any constraints on the neural networks (other than small $l_1$ regularization).
>
> Since we estimate the Lipschitz constant of the two neural networks at each iteration during training, an efficient method for this estimation is crucial. Existing methods, such as those proposed by Wang et al., are either computationally expensive or too restrictive. Furthermore, the method introduced by Pracht et al. is only applicable to linear neural networks.
>
> We acknowledge that other approaches in the literature provide tighter bounds on the Lipschitz constant compared to Combettes and Pesquet, such as Fazlyab et al., "Efficient and accurate estimation of Lipschitz constants for deep neural networks," NeurIPS 2019, and Pauli et al., "Novel quadratic constraints for extending LipSDP beyond slope-restricted activation," ICLR 2024. However, these methods are computationally expensive. In fact, our numerical experiments indicate that approximately $99\\%$ of the training time is spent calculating the Lipschitz constant, with only$  1\\% $ dedicated to actual training.
>
> Instead, we compute an upper bound of the Lipschitz constant using the spectral norm. For details, we refer the reviewer to "Lipschitz certificates for layered network structures driven by averaged activation operators," SIAM Journal on Mathematics of Data Science, by Combettes and Pesquet. The spectral norm approach offers a good trade-off: it provides a much tighter bound than the trivial upper bound while remaining efficient to compute.
>
> We performed an a posteriori comparison between the spectral norm approach and the method proposed by Fazlyab et al. Our results show that the spectral norm approach is an order of magnitude faster than Fazlyab et al., while the upper bound it provides is only $40\\%$  to  $50\\%$ larger than the upper bound obtained by Fazlyab. This finding aligns with the results reported in Fazlyab et al., particularly in Figure 2a.
>
> We sincerely thank the reviewer for their detailed analysis and thoughtful feedback on our method. We appreciate the time and effort you have dedicated to reviewing our paper and will incorporate your suggestions to enhance its clarity and quality.
>
> We kindly request that you re-evaluate our paper in light of the clarifications and additional insights we have provided.
>
> Sincerely,
>
> Authors

---

> > ### Comment · Reviewer_RLbj · 2024-11-27
> >
> > Thank you for your careful clarification. I think the approach fills a gap among current frameworks for certifying+synthesizing learned controllers, but I still have concerns about the scalability and relevance to the ICLR community. I have raised my score accordingly.
> >
> > Regarding the loss being zero, I did not mean to say that its required for the validity of the arguments in the paper. My point is to say that if some other control objective is added to the controller loss, the MSE barrier certificate loss will always be an "active" constraint which competes with the performance objective (whatever the user defines them to be). It is very uncommon in practice to formulate a control problem only by forward set invariance without specifying any other objectives like control effort or some tracking error. It's interesting that the controller you synthesized happen to converge to an equilibrium point, but it seems like this is not a totally predictable behavior.
> >
> > I think it worth mentioning some related papers that work on learning control barrier functions from data in the continuous-time setting. They also use similar Lipschitz and epsilon-net covering arguments.
> >
> > (Robey 2020 CDC) Learning Control Barrier Functions from Expert Demonstrations
> > (Lindemann 2021 CoRL) Learning Hybrid Control Barrier Functions from Data
> > (Lindemann  2021 IEEE) Learning Robust Output Control Barrier Functions from Safe Expert Demonstrations
> >
> > Also there is some recent work that certifies regions of attractions (also forward invariant sets) using automatic neural verification tools like alpha-beta CROWN. This approach does not require any explicit Lipschitz constant estimation, but limited by poor scaling of branch-and-bound algorithms.
> >
> > (Yang 2024, ICML)  Lyapunov-stable Neural Control for State and Output Feedback: A Novel Formulation

---

> > > ### Author Response · Authors · 2024-11-27
> > >
> > > We thank the reviewer for new suggestions and raising their score.
> > >
> > > Q. I still have concerns about the scalability and relevance to the ICLR community. I have raised my score accordingly.
> > >
> > > A. Unfortunately, methods that provide formal guarantees do not scale well, as we highlighted in our previous comment. We believe our work is highly relevant to the ICLR community, as, to the best of our knowledge, it is the first to address complex systems like the double inverted pendulum. As noted earlier, even model-based approaches have been unable to identify a control barrier certificate for this system.
> > >
> > > We acknowledge that our method is limited by exponential sample complexity. To address this, we have added a new section on limitations, where this issue is discussed in detail.
> > >
> > > Q. Regarding the loss being zero, I did not mean to say that its required for the validity of the arguments in the paper. My point is to say that if some other control objective is added to the controller loss, the MSE barrier certificate loss will always be an "active" constraint which competes with the performance objective (whatever the user defines them to be). It is very uncommon in practice to formulate a control problem only by forward set invariance without specifying any other objectives like control effort or some tracking error. It's interesting that the controller you synthesized happen to converge to an equilibrium point, but it seems like this is not a totally predictable behavior.
> > >
> > > A. Thank you for the clarification. During training, points that satisfy the conditions of the barrier certificate do not influence the training process. We refer the reviewer to our code (specifically, the "check" function) for further details. Additionally, we have incorporated a small loss term in our implementation to encourage states to reach zero, ensuring that the system converges to an equilibrium point. Based on our experiments, this loss term does not affect the overall convergence.
> > >
> > >
> > > We have incorporated reviewer's suggestions in our paper, and we have cited those papers accordingly.

---

### Author Response · Authors · 2024-12-02
**Author Rebuttal**

We would like to thank the reviewers for providing detailed comments that have helped to improve the quality of our manuscript. We have provided rebuttals to the comments of each reviewer.

Summary of changes:

1. Justification for using MSE loss:

We have added a new paragraph explaining why the MSE loss theoretically results in a smaller Lipschitz constant compared to ReLU loss. This provides additional clarity on the benefits of our approach.

2. Justification for using Spectral norm to estimate the Lipschitz upper bound of a neural network:

We acknowledge that other approaches in the literature provide tighter bounds on the Lipschitz constant compared to Combettes and Pesquet, such as Fazlyab et al., "Efficient and accurate estimation of Lipschitz constants for deep neural networks," NeurIPS 2019, and Pauli et al., "Novel quadratic constraints for extending LipSDP beyond slope-restricted activation," ICLR 2024. However, these methods are computationally expensive. In fact, our numerical experiments indicate that approximately $99\\%$ of the training time is spent calculating the Lipschitz constant, with only$  1\\% $ dedicated to actual training.

Instead, we compute an upper bound of the Lipschitz constant using the spectral norm. For details, we refer the reviewer to "Lipschitz certificates for layered network structures driven by averaged activation operators," SIAM Journal on Mathematics of Data Science, by Combettes and Pesquet. The spectral norm approach offers a good trade-off: it provides a much tighter bound than the trivial upper bound while remaining efficient to compute.

We performed an a posteriori comparison between the spectral norm approach and the method proposed by Fazlyab et al. Our results show that the spectral norm approach is an order of magnitude faster than Fazlyab et al., while the upper bound it provides is only $40\\%$  to  $50\\%$ larger than the upper bound obtained by Fazlyab. This finding aligns with the results reported in Fazlyab et al., particularly in Figure 2a.

3.  Limitations Section:

In response to reviewer feedback, we have added a dedicated limitations section that clearly outlines the shortcomings of our method.

---

### Meta-Review · Area_Chair_5CL5 · 2024-12-21

**Metareview:**

This paper proposes replacing the traditional ReLU based loss with MSE loss to synthesize control barrier functions. The authors shows in various experiments that the proposed method is effective. There are some mixed opinions among the reviewers regarding the presentation and clarity of the paper, and a commonly shared concerns on the scalability, gridding approach, and calculation of Lipschitz constant. Using sampling based approach for estimation as suggested by the authors seem to destroy the guarantee unless it provide an upper bound. Overall, I think this paper is at the boundary of reject/acceptance, and the authors are encouraged to incorporate the reviewers suggestions including both technical points and writing suggestions in the revision.

**Additional Comments On Reviewer Discussion:**

There are a few shared major questions regarding the use of loose lipschitz bound and the missing connection to recent literature on tighter lipschitz bound in the neural network verification literature. The authors responded that they used a less tight bound due to computation concern and compared with one SDP approach (Fazlyab etal) showing that tighter bound is more computationally expensive. However, there are other approach in the neural network verification literature that can provide slightly less tight bound using SDP but being much more computationally efficient (e.g. using convex relaxation approach, using linear bounding approach like Crown, or using interval bound propagation like IBP). It is recommended that the authors investigate further into the connections which will further strengthen the paper.

---

### Decision · Program_Chairs · 2025-01-22

Reject